# Characterization of SARS-CoV-2 and host entry factors distribution in a COVID-19 autopsy series

Xiao-Ming Wang[1,2,6], Rahul Mannan [1,2,6], Lanbo Xiao[1,2], Eman Abdulfatah[1], Yuanyuan Qiao[1,2], Carol Farver[1], Jeffrey L. Myers[1], Sylvia Zelenka-Wang[1,2], Lisa McMurry[1], Fengyun Su[2], Rui Wang[2], Liron Pantanowitz[1], Jeffrey Jentzen[1], Allecia Wilson[1], Yuping Zhang[2], Xuhong Cao[2], Arul M. Chinnaiyan[1,2,3,4,5,6] & Rohit Mehra [1,2,3,6✉]

## Abstract

**Background** SARS-CoV-2 is a highly contagious virus that causes the disease COVID-19. We have recently reported that androgens regulate the expression of SARS-CoV-2 host entry factors ACE2 and TMPRSS2, and androgen receptor (AR) in lung epithelial cells. We also demonstrated that the transcriptional repression of the AR enhanceosome inhibited SARS-CoV-2 infection in vitro.

**Methods** To better understand the various sites of SARS-CoV-2 infection, and presence of host entry factors, we extensively characterized the tissue distribution and localization of SARS-CoV-2 virus, viral replication, and host entry factors in various anatomical sites sampled via autopsy. We applied RNA *in-situ*-hybridization (RNA-ISH), immunohistochemistry (IHC) and quantitative reverse transcription polymerase chain reaction (qRT-PCR) approaches. We also assessed histopathological changes in SARS-CoV-2 infected tissues.

**Results** We detect SARS-CoV-2 virus and viral replication in pulmonary tissues by RNA-ISH and IHC and a variety of non-pulmonary tissues including kidney, heart, liver, spleen, thyroid, lymph node, prostate, uterus, and colon by qRT-PCR. We observe heterogeneity in viral load and viral cytopathic effects among various organ systems, between individuals and within the same patient. In a patient with a history of kidney transplant and under immunosuppressant therapy, we observe an unusually high viral load in lung tissue by RNA-ISH, IHC and qRT-PCR. SARS-CoV-2 virus is also detected in this patent's kidney, liver and uterus. We find *ACE2*, *TMPRSS2* and *AR* expression to overlap with the infection sites.

**Conclusions** This study portrays the impact of dispersed SARS-CoV-2 infection in diverse organ systems, thereby facilitating avenues for systematic therapeutic approaches.

## Plain language summary

To understand SARS-CoV-2 infection of human organs, we characterized the tissue distribution of SARS-CoV-2 virus, and the presence of host factors that enable the virus to enter cells, in postmortem tissues from six patients who had COVID-19. We assessed the presence of SARS-CoV-2 viral RNA and the expression of human genes that facilitate virus entry in host cells, using several techniques. We observed that SARS-CoV-2, and factors that facilitate virus entry in host cells, were present in the same location in pulmonary and multiple non-pulmonary tissues, including lung, bronchus, trachea, kidney, heart, liver, spleen, thyroid, lymph node, prostate, uterus, and colon. We also reported changes in the microscopic appearance of SARS-CoV-2 infected tissues at various sites. Such findings will guide future coronavirus biology studies on patients with advanced disease.

[1] Department of Pathology, University of Michigan Medical School, Ann Arbor, MI, USA. [2] Michigan Center for Translational Pathology, Ann Arbor, MI, USA. [3] Rogel Cancer Center, Michigan Medicine, Ann Arbor, MI, USA. [4] Department of Urology, University of Michigan Medical School, Ann Arbor, MI, USA. [5] Howard Hughes Medical Institute, Ann Arbor, MI, USA. [6] These authors contributed equally: Xiao-Ming Wang, Rahul Mannan, Arul M. Chinnaiyan, Rohit Mehra. ✉email: mrohit@med.umich.edu

Coronavirus disease-19 (COVID-19), an infectious disease caused by a novel coronavirus called severe acute respiratory syndrome coronavirus 2 (SARS-CoV-2), was declared a pandemic by the World Health Organization on March 11, 2020. As of May 21st, 2021, 165,874,001 diagnosed cases and 3,438,383 deaths have been reported worldwide (https://coronavirus.jhu.edu/), with the United States of America bearing the highest disease impact in terms of morbidity and mortality (33,084,872 cases and 589,222 deaths) followed by India where is currently undergoing a massive surge of cases. COVID-19 is an established multiorgan disease in humans with the greatest involvement and derangement involving the respiratory, cardiovascular, renal, and immune systems[1].

Coronaviruses are a group of enveloped viruses with a single-stranded RNA genome. In rare instances, animal coronaviruses such as severe acute respiratory syndrome (SARS) coronavirus (SARS-CoV) in 2002, Middle East respiratory syndrome (MERS) coronavirus (MERS-CoV) in 2012, and currently SARS-CoV-2, can infect humans with variable clinical complications and impact. All three of these coronaviruses are transmitted zoonotically and spread among humans through close contact[2]. SARS-CoV-2 is most commonly transmitted from person to person via respiratory droplets. Most patients experience mild symptoms. However, a significant subset of patients may experience severe disease outcomes including long term health sequelae and death[3]. The SARS-CoV-2 genome shares ~80% sequence identity with SARS-CoV and MERS-CoV, although the sequence similarity varies among genes encoding structural proteins and essential enzymes. SARS-CoV-2 is considered to be more pathogenic than the MERS-CoV and SARS-CoV[4,5]. Structurally, SARS-CoV-2 is comprised of four structural proteins: spike (S), envelope (E), membrane glycoprotein (M), and nucleo-capsid phosphoprotein (N) proteins[5]. SARS-CoV-2 entry into host cells depends on binding of viral spike proteins to the host receptor protein angiotensin-converting enzyme 2 (ACE2) and priming by the host serine protease, the cell surface transmembrane protease serine 2 (TMPRSS2)[6,7]. In our recent study detailing transcriptional regulation of SARS-CoV-2 entry factors ACE2 and TMPRSS2, we described co-expression of the androgen receptor (AR), TMPRSS2, and ACE2 in bronchial and alveolar cells in human and murine lung tissues; demonstrated that androgens regulate the expression of ACE2, TMPRSS2, and AR in subpopulations of lung epithelial cells; and reported that transcriptional repression of the AR enhanceosome by AR antagonists inhibited SARS-CoV2 infection in vitro[8]. To further investigate transcriptional inhibition of critical host factors in the treatment or prevention of COVID-19, we set out to characterize the distribution and localization of SARS-CoV-2 viral particles, viral replication, and the host entry factor receptor ACE2, priming protease TMPRSS2 and transcriptional regulator AR by RNA in situ hybridization (RNA-ISH), immunohistochemistry (IHC) and qRT-PCR in various pulmonary and nonpulmonary tissues obtained from a series of six clinical autopsies performed on patients who succumbed to COVID-19 disease. We also correlated the presence of SARS-CoV-2 viruses and host entry machinery with clinicopathologic findings in the affected organ systems. We detect SARS-CoV-2 viral RNA and viral replication events in both pulmonary tissues and nonpulmonary tissues. We observe heterogeneity in viral load and viral cytopathic effects among various organ systems, between individuals and within the same patient. We also find the presence of host factors, ACE2, TMPRSS2, and AR, to overlap with the infection sites.

## Methods

**Patient selection**. This study was performed under ethical approval protocols of the Institutional Review Boards of the University of Michigan Medical School (IRBMED). Six patients with SARS-CoV-2 infection who died from COVID-19 disease with clinical autopsies performed at Michigan Medicine were included in the study. The legal next of kin of patients provided the informed consent prior to autopsy. All the patients tested positive for SARS-CoV-2 by quantitative reverse transcription polymerase chain reaction (qRT-PCR) performed in a clinical laboratory. This study was conducted prior to the FDA approval of emergency use of COVID-19 vaccines and none of the patients in this study received COVID-19 vaccination. During autopsy, multiple tissue samples were harvested from various pulmonary and nonpulmonary sites for clinicopathologic characterization. For the purposes of this study, representative FFPE tissue blocks (n = 74) were selected from 16 different anatomical sites by two pathologists (R. Mannan and R. Mehra) based on the evaluation of hematoxylin and eosin (H&E) stained FFPE slides with brightfield microscopy.

**Histopathological evaluation**. The pathological appraisal of tissue samples was performed for pulmonary and nonpulmonary organs. The histopathological assessment for the respiratory region was carried out on the trachea, bronchus, and pulmonary parenchymal tissues. For nonrespiratory sites, we evaluated myocardial tissues, hematological tissues (lymph node and spleen), liver, and tissues from the gastrointestinal (esophagus, stomach, small intestine, and colon) tract, endocrine (thyroid, pancreas, and adrenal) glands, and genitourinary (kidney, prostate, and uterus) tract. A systematic evaluation for histo-morphological and pathological changes were noted and recorded for each sample.

**RNA in situ hybridization (RNA-ISH)**. RNA-ISH was performed on 4 μm FFPE tissue sections using RNAscope 2.5 HD Brown kit, duplex kit and target probes against SARS-CoV-2 spike (S) gene (SARS-CoV-2-S probe), minus strand of SARS-CoV-2-S gene (SARS-CoV-2-S-sense), SARS-CoV-2 nucleocapsid (N) gene (SARS-CoV-2-N-O1-C2), human ACE2, TMPRSS2, and AR genes (Supplementary Table 1) (Advanced Cell Diagnostics, Newark, CA). The SARS-CoV-2-S-sense probe detects the minus strand viral RNA, which is present during active viral replication. RNA quality was assessed using Hs-PPIB probe as positive control and assay background was monitored using DapB probe as negative control. In addition, tissue sections from one normal lung and one H1N1 influenza patient lung were used as negative control samples for SARS-CoV-2 probes. RNA-ISH assay was performed as previously described[9–11]. After deparaffinization, FFPE sections were pretreated with hydrogen peroxide followed by heat-induced target retrieval and protease, and subsequently hybridized with target probe followed by a series of signal replications. Finally, chromogenic detection was performed using DAB and counterstained with 50% Gill's hematoxylin I (Fisher Scientific, Rochester, NY). To confirm that the RNA-ISH signals were amplified from viral RNA, tissue sections were treated with 5 mg/ml RNase A for 30 min at 40 °C prior to target probe hybridization.

**Immunohistochemistry (IHC)**. IHC was performed on 4 μm FFPE tissue sections on the Ventana Discovery XT automated slide staining system (Roche-Ventana, AZ) using the Chromo-Map diaminobenzidine (DAB) detection kit (760-500, Roche-Ventana, AZ). SARS-CoV-2 nucleocapsid IHC was performed using mouse monoclonal antibody (R&D system MAB10474) at 1:100 dilutions with heat-induced epitope retrieval with TRIS antigen retrieval buffer.

**RNA extraction and quantitative RT-PCR (qRT-PCR)**. Total RNA was extracted from two 5 µm FFPE tissue sections using the miRNeasy FFPE Kit (Qiagen, Hilden, Germany) according to the manufacturer's protocol and eluted into 30 µl H2O.

For SARS-CoV-2 RNA detection, 8 µl RNA template was used per qRT-PCR reaction using the GenePath CoViDx One kit (Maharashtra, India), which contains premixed RT-PCR primer sets for SARS-CoV-2 Envelop (E) gene, Nucleocapsid (N) gene, RNA-dependent RNA polymerase (RdRP) gene and the human RNaseP, an internal extraction and amplification control. Samples were classified as SARS-CoV-2 positive when at least one of the three SARS-CoV-2 genes were detected with a Ct value < 40. Each reaction was performed in replicates. Tissues from one normal lung, one H1N1 influenza patient lung and one normal prostate were selected as negative control. For undetermined data points, Ct value was set to 40 to calculate the ΔCt.

For viral replication detection, we performed two-step strand-specific RT-PCR as previously described[12]. Frist, strand-specific primers were used to reverse transcribe SARS-CoV-2 viral RNA to cDNA in two separate reactions: a forward E gene primer to generate minus strand cDNA representing viral replication and a reverse E gene primer to generate plus strand cDNA representing the single-strand virus. In addition, a separate set of human RNaseP primers were added to each set of reactions to serve as internal amplification control. Second, each cDNA sample was amplified by real-time PCR to detect either SARS-CoV-2 E gene or human RNaseP gene by specific primer sets using SYBR Green (Bio-Rad, Hercules, CA).

The RT-PCR data are available in the supplementary information files. All other data that support the findings of this study are available from the corresponding author upon reasonable request.

**Reporting summary**. Further information on research design is available in the Nature Research Reporting Summary linked to this article.

## Results

**Clinical and histopathologic features**. The study cohort was comprised of five males and one female patient with age ranging from 37 to 80 years old (median age of 59.5 years). The most common clinical presentations included pyrexia (5/6), followed by acute hypoxemic failure refractory to supplemental oxygen (4/6), cough and dyspnea (2/6), and diarrhea (1/6). All six patients had pre-existing co-morbidities that included a history of diabetes and hypertension (5/6 patients), asthma (2/6), obesity (1/6), hypercholesterolemia (1/6), coronary artery disease (CAD) (1/6), and renal transplantation (1/6). All patients exhibited abnormal pulmonary findings on radiologic imaging including the presence of ground-glass opacities on computed tomography (CT). The cause of death for all six patients was attributed to pulmonary complications associated with SARS-CoV-2 infection recorded to be due to acute respiratory distress syndrome (ARDS) in 5/6 patients and bronchopneumonia in one patient.

**Pulmonary histopathologic findings**. Histologic examination of lung tissues revealed acute diffuse alveolar damage (DAD) with prominent hyaline membranes in 5/6 patients (Fig. 1a) and diffuse lymphocytic inflammatory infiltrates in all six patients. Evidence of coagulopathy (i.e., microthrombi, capillaritis etc.) was identified in a majority of patients (5/6). Additional histopathological findings attributed to viral cytopathic effects include pronounced usual hyperplastic alveolar pneumocytes (2/6 patients) and intracytoplasmic vacuolation (1/6 patient). Another histopathological feature observed in the cohort, possibly not

directly attributed to COVID-19, was bronchopneumonia (3/6 patients). One patient with underlying asthma showed associated changes including mucus plugging, goblet cell metaplasia, mucus gland hyperplasia, and thickening of subepithelial basement membranes, while another patient with chronic obstructive pulmonary disease (COPD) showed large emphysematous changes and edema. A summary of pulmonary histopathologic findings is presented in Supplementary Table 2.

**Nonpulmonary histopathologic findings**. Histologic examination of the nonpulmonary organs revealed notable pathology affecting the myocardium (5/6 patients), kidney (2/4) and liver (3/6). A summary of nonpulmonary histopathologic findings is presented in Supplementary Table 2. Myofibrillary hypertrophy was identified in cardiac tissues (5/6 patients) with evidence of focal to diffuse fibrosis in a subset of cases. Evidence of chronic inflammation was not only identified in half of the cases, mostly in the pericardial region, but also observed in the perivascular region in one patient. Renal sampling was performed in 4/6 patients with histopathological changes mostly observed in the medullary region of the kidney where tubules were affected. Signs of tubular injury include tubular epithelial sloughing within lumina admixed with red blood cells and calcification. Patient 3, who had a unique clinical history of diabetic nephropathy status post-transplant, had a focal collection of large, multinucleated, dyscohesive cells with foamy cytoplasm in the renal medulla of their transplanted kidney and histological changes consistent with end-stage renal disease (ESRD) in the native kidneys. Histological examination of liver revealed mild to moderate steatosis in all patients where liver tissue was sampled. However, no evidence of overt inflammation or fibrosis was observed.

**Detection of SARS-CoV-2 virus in pulmonary tissues by RNA-ISH**. We performed RNA-ISH using a target probe against the SARS-CoV-2-S gene for detection of viral particles in formalin fixed paraffin embedded (FFPE) tissue sections. The SARS-CoV-2 RNA-ISH signals were identified as individual brown punctate dots or clusters. As expected, SARS-CoV-2 viral particles were detected in lung tissues of all six patients and no signal was observed in normal lung tissue or H1N1 influenza patient's lung tissue. The viral loads, however, varied drastically among patients and different sampling sites from the same organ of a single patient. RNA-ISH signals were observed to be patchy. At anatomic level, the SARS-CoV-2 viral particles were observed in the intact as well as desquamated alveolar epithelial cells lying in the edema fluid of the intra-alveolar space and within the hyaline membranes lining the alveoli. (Fig. 1a–d). SARS-CoV-2 virus was also detected in the nonalveolar region such as peri-bronchial sero-mucinous glands in 2/4 patients (Fig. 1e, f) and peri-tracheal region in 1/4 patients, but with lower viral counts compared to the alveolar region as indicated by RNA-ISH staining. Viral signals in lung tissue were also noted to be prominent in areas exhibiting necrosis and sero-fibrinous deposits. (Fig. 1g, h).

As described in the methods, we also employed a sense probe against the minus strand of SARS-CoV-2-S gene to detect viral replication. Utilizing this approach, we observed active viral replication events as brown signal dots and clusters within the intra-alveolar fibrinous deposits, the cellular proliferation seen in the alveolar septa and bronchiolar region of lung parenchyma in all six patients. Signals were also identified in primary/parent bronchus tissues in 2/4 patients, and tracheal tissues in 1/4 patients (Fig. 2).

To demonstrate and confirm the specificity of the virus signals detected by RNA-ISH described above, we performed duplex RNA-ISH assay for co-detection of SARS-CoV-2-N gene and S

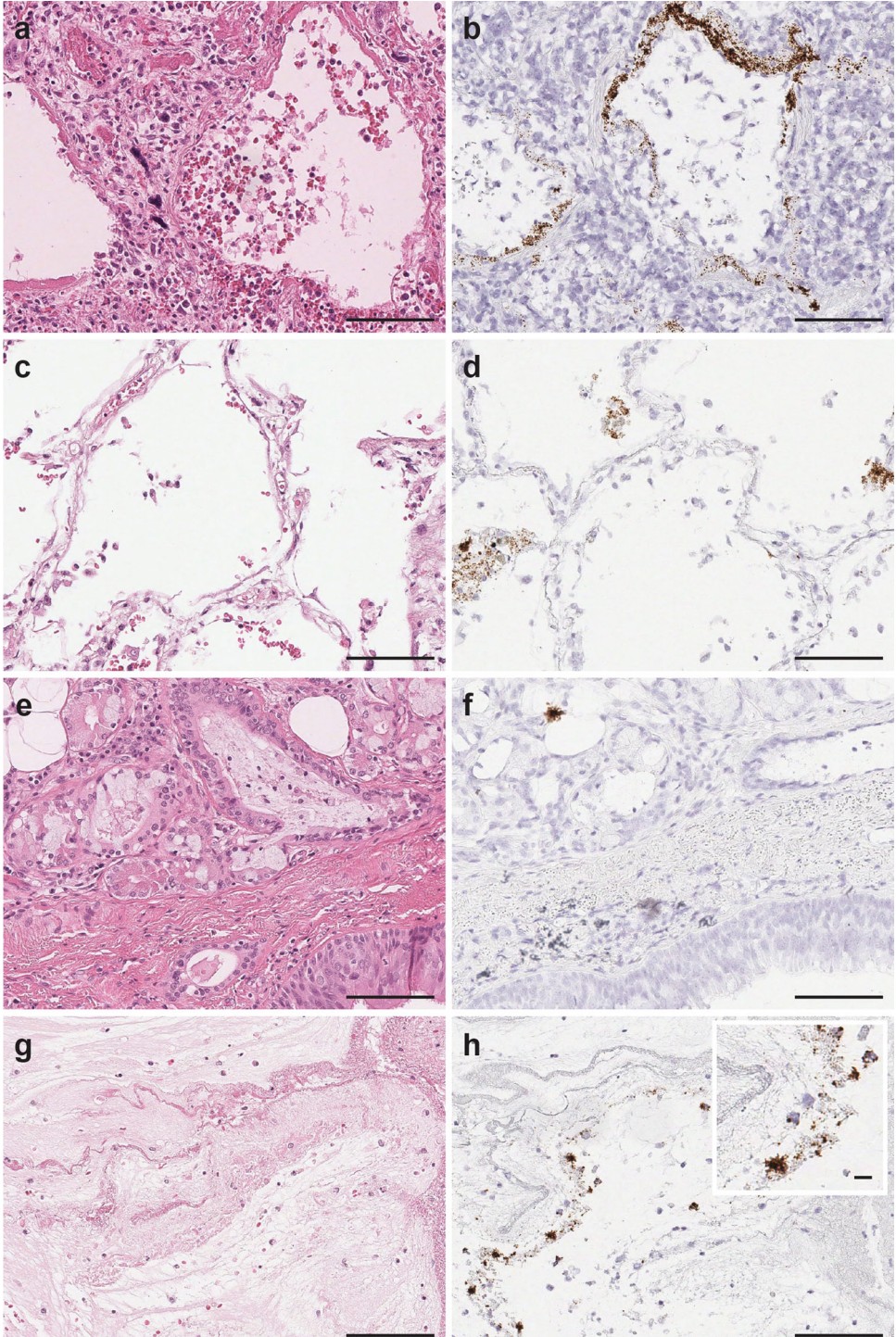

**Fig. 1 SARS-CoV-2 virus detection in pulmonary parenchyma by SARS-CoV-2 spike RNA-ISH.** The viral signals were observed within the intra-alveolar hyaline membranes (**a** H&E and **b** SARS-CoV-2 spike RNA-ISH) and within the intact lining alveolar epithelial cells as well as in the desquamated alveolar epithelial cells (**c** H&E and **d** SARS-CoV-2 spike RNA-ISH). Viral particles were also noted in the nonalveolar region like bronchus as a cluster of viral signals within peri-bronchial sero-mucinous glands, as well as within the lining pseudostratified respiratory bronchial epithelium. (**e** H&E and **f** SARS-SoV-2 spike RNA-ISH). Viral signals were also detected in the necrotic and fibrinous material within damaged pulmonary tissue (**g** H&E and **h** SARS-CoV-2 spike RNA-ISH). Inset: Viral signals shown as individual punctate brown dots and clusters. Scale bars = 100 μm. Inset scale bar = 10 μm.

gene in the lung autopsy tissue sections. We observed co-presence of both N gene and S gene signals within the same topographical locations of intra-alveolar region, hyaline membranes and peri-tracheal sero-mucinous glands (Fig. 3a, b). Similar strategy was utilized by combining the target probes against N gene and the minus strand of SARS-CoV-2-S gene, to show case SARS-CoV-2

viral replication event co-existing within the same cell types harboring the virus genome (Fig. 3c, d).

As an additional confirmatory validation for RNA-ISH signal specificity, we treated the autopsy tissue sections with RNase. Upon RNase treatment just prior to SARS-CoV-2 target probe hybridization, RNA-ISH signals were no longer detectable within

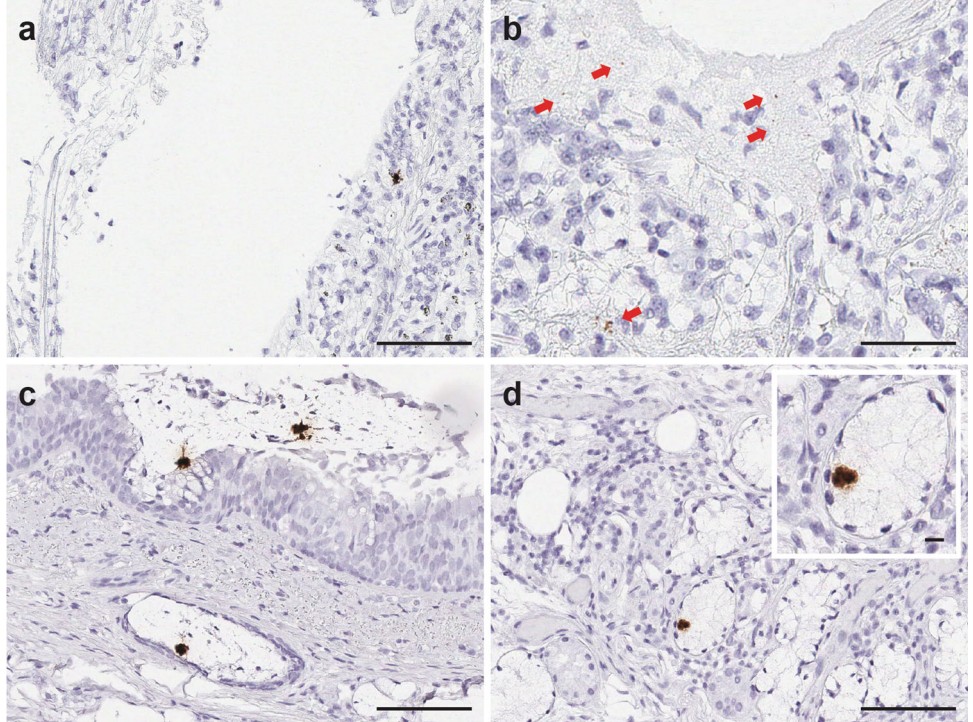

**Fig. 2 Detection of SARS-CoV-2 viral replication in pulmonary parenchyma by SARS-CoV-2-S-sense RNA-ISH.** SARS-CoV-2 viral replication events were observed in the intra-alveolar septal region (**a**), hyaline membranes lining the alveolar space (**b**), ciliated columnar respiratory epithelium of the bronchus (**c**), and the subepithelial sero-mucinous tracheal glands (**d**). Inset: a signal cluster representing viral replication. Scale bars = 200 μm in **a**, 100 microns in **b**–**d**. Inset scale bar = 10 μm. Arrows point to RNA-ISH signals.

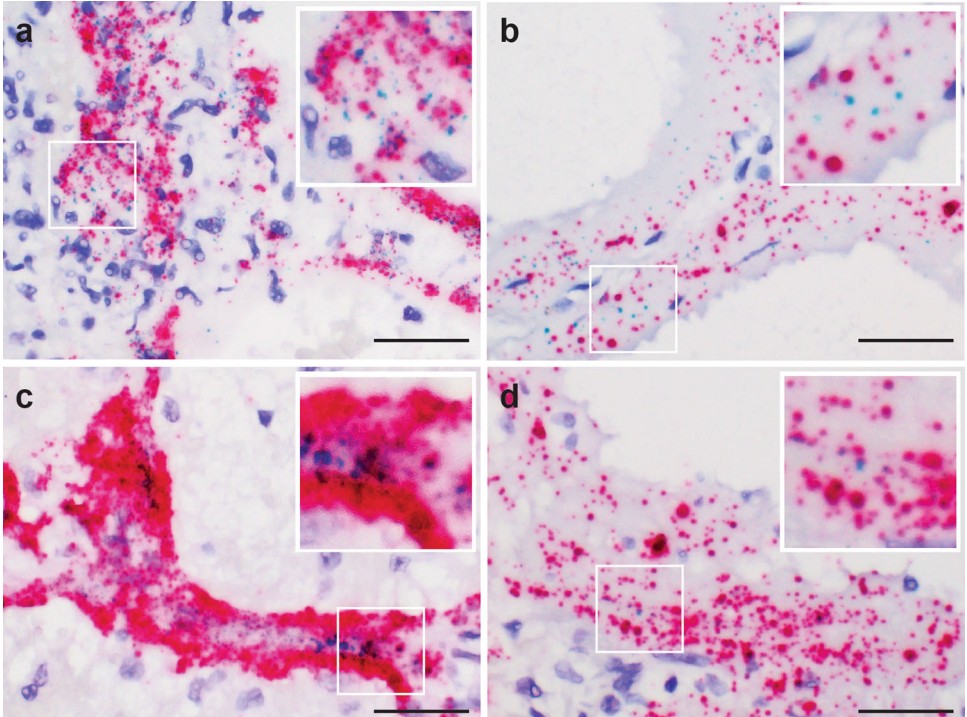

**Fig. 3 Co-detection of SARS-CoV-2 virus spike (S), nucleocapsid (N) genes, and viral replication in pulmonary parenchyma.** Co-expression of the SARS-CoV-2-S (green) and N (red) genes in the intra-alveolar (**a**) and hyaline membranes in lung parenchyma (**b**) utilizing the duplex RNA-ISH with probes against the SARS-CoV-2-N gene and -S gene. Co-detection of the SARS-CoV-2 virus (red) and viral replication (green) in the intra-alveolar and hyaline membranes in lung parenchyma (**c**, **d**) with probes against the SARS-CoV-2-N gene and minus strand of S gene. (**a**, **c** from patient 1, **b, d** from patient 3) Scale bars = 50 μm.

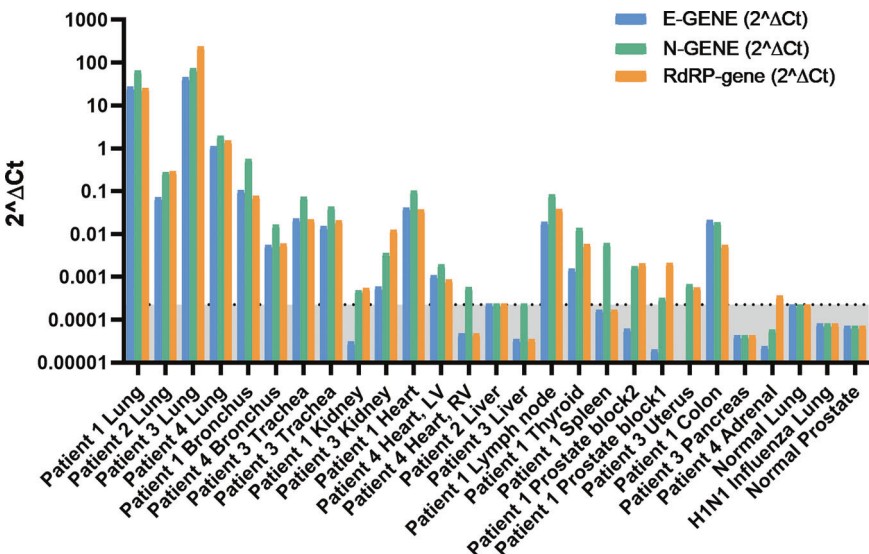

**Fig. 4 qRT-PCR of SARS-CoV-2 infection in COVID-19 autopsy tissues.** $2^{(\Delta Ct)}$ for SARS-CoV-2 E gene, N gene, and RdRP gene using RNaseP housekeeping gene as reference. The undetermined Ct was set to 40. Normal lung, H1N1 influenza lung, and normal prostate tissues were used as negative controls. The $2^{(\Delta Ct)}$ of normal lung was set as a cutoff point (0.000224) and the gray area below the cutoff line is considered as SARS-CoV-2 negative. Minimum 1 out of the 3 genes should have Ct value < 40 (or $2^{(\Delta Ct)} > 0.000224$) to be considered as SARS-CoV-2 positive.

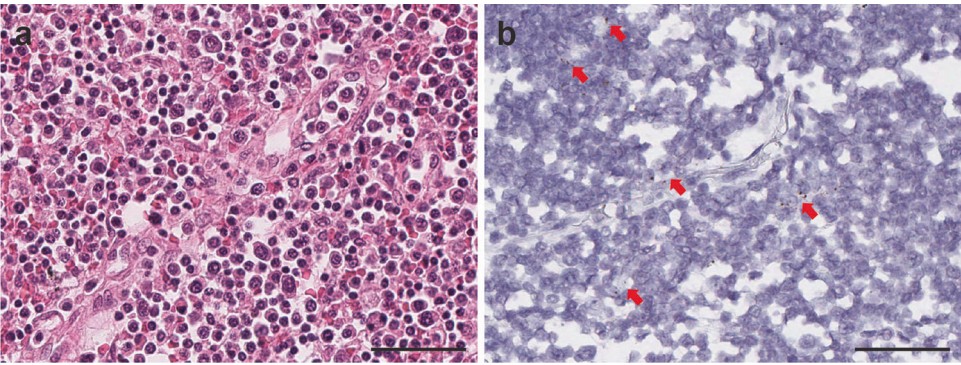

**Fig. 5 SARS-CoV-2 virus detection within lymph node.** Viral signal clusters were observed within lymph node germinal center as individual brown dots (**a** H&E and **b** SARS-CoV-2 RNA-ISH, in arrow pointed area). Scale bars = 50 µm.

the tissue samples, which were earlier reported to be SARS-CoV-2 positive by conventional RNA-ISH (Supplementary Fig. 1).

**Detection of SARS-CoV-2 virus in pulmonary tissues by IHC.** We found a concordance between SARS-CoV-2 nucleocapsid IHC positivity and SARS-CoV-2 RNA-ISH positivity in the pulmonary autopsy tissue samples. The IHC expression was consistent with RNA-ISH assay in terms of pattern and topographical localization as it was also observed in the hyaline membrane and intra-alveolar region (Supplementary Fig. 2).

**Detection of SARS-CoV-2 virus in pulmonary tissues by qRT-PCR.** We also performed qRT-PCR on eight SARS-CoV-2 RNA-ISH-positive pulmonary tissue samples from four patients using primer sets for E gene, N gene, and RdRP gene. All the selected pulmonary tissues were also qRT-PCR positive (Fig. 4). On the other hand, none of the genes were detected in normal lung tissue or H1N1 influenza patient's lung tissue. The SARS-CoV-2 RNA-ISH results were highly consistent with the qRT-PCR results with an overall good correlation of signal intensities among pulmonary tissues (Supplementary Table 3).

To further confirm the RNA-ISH signals of viral replication, we performed minus strand-specific reverse transcription followed

by E gene real-time PCR for lung samples positive for viral replication as detected by RNA-ISH. The PCR results are consistent with the RNA-ISH results for both S-sense RNA-ISH-positive and -negative samples. (Supplementary Table 4)

**Detection of SARS-CoV-2 virus in nonpulmonary tissues.** To further investigate the SARS-CoV-2 tissue distribution, we selected 14 representative nonpulmonary tissue samples, including kidney, heart, liver, lymph node, spleen, thyroid, prostate, uterus, colon, adrenal, pancreas, stomach, esophagus, and small intestine, using RNA-ISH and qRT-PCR methodologies. Low level signals with high cycle threshold (Ct) values were detected in kidney, heart, liver, lymph node, thyroid, spleen, prostate, uterus, colon, and adrenal by qRT-PCR. (Fig. 4 and Supplementary Table 3). Rare RNA-ISH signal dots were observed within the germinal center of the cortical lymphoid follicle of a lymph node sample from Patient 1 (Fig. 5) and within the distal tubules of a kidney sample form Patient 7 (Fig. 6c, d). Both tissues were qRT-PCR positive (Fig. 4, Supplementary Table 3, Patient 1 Lymph node with average Ct 30.8 and Patient 3 Kidney with average Ct 36.5). Very rare signal clusters were observed in kidney, heart, liver, thyroid, spleen, prostate, uterus, and colon tissue sections, but no individual signal dots representing viral particles were

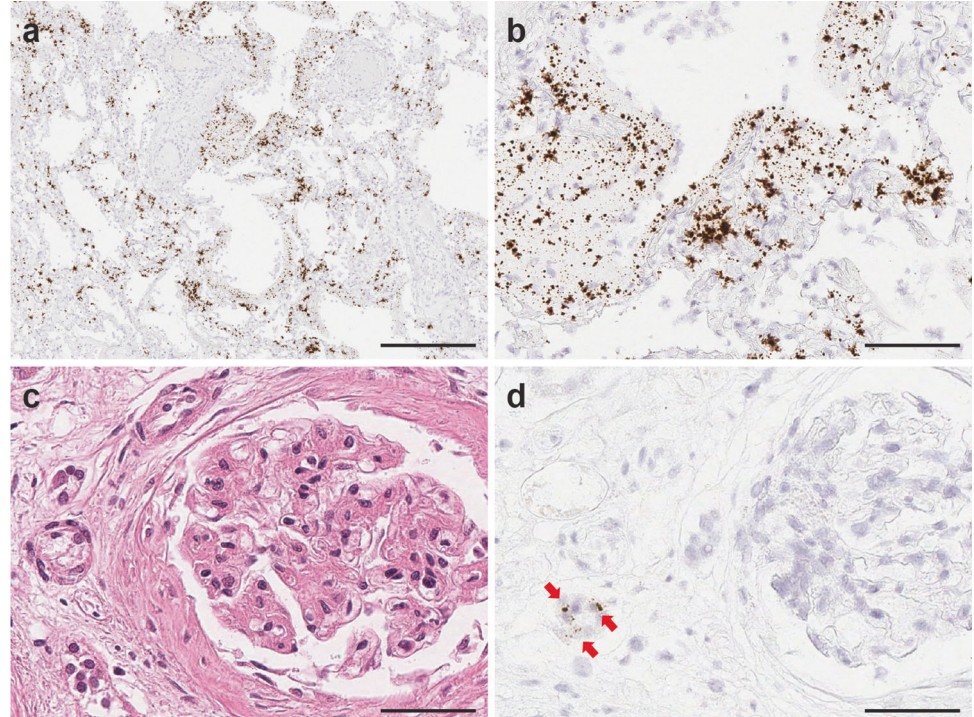

**Fig. 6 SARS-CoV-2 infection in kidney transplant patient (patient 3) on immunosuppressant therapy.** Highly abundant viral signals were observed within the lung parenchyma exhibiting diffuse alveolar damage, reflecting the high SARS-CoV-2 virus infection load in Patient 3. The viral signals were detected in intra-alveolar hyaline membrane and interstitial fibroblastic proliferation region (**a, b**. SARS-CoV-2 RNA-ISH). Scattered viral signals were identified in the renal tubules (**c** H&E and **d** SARS-CoV-2 RNA-ISH). Scale bar = 500 μm in **a**, 100 μm in **b**, 50 μm in **c, d**.

detected in any of those tissues. Therefore, we did not consider these tissues as SARS-CoV-2 positive by RNA-ISH.

**High abundance of SARS-CoV-2 viruses found in lung tissues of a patient status post kidney transplant**. Patient 3 in our clinical autopsy series had a clinical history of kidney transplant for end-stage diabetic nephropathy and was receiving immunosuppressant therapy with mycophenolate and prednisolone. SARS-CoV-2 RNA-ISH staining revealed very high viral infection in the hyaline membranes and intra-alveolar septum throughout the lung tissue from this immunosuppressed patient (Fig. 6a, b). Viral inflammatory changes were observed in the transplant kidney and RNA-ISH indicated SARS-CoV-2 viral signals within the distal tubules of the transplant kidney (Fig. 6c, d). SARS-CoV-2 viral signals were also detected in the liver and uterus tissues by qRT-PCR (Supplementary Table 3).

**Localization of ACE2, TMRPSS2, and AR expression in various organ systems**. ACE2 and TMPRSS2 RNA-ISH signals in pulmonary tissues were noted in tracheal epithelial cells in 2/2 patients, peri-bronchial glands in 2/4 patients, bronchial and bronchiolar respiratory epithelial cells and a subpopulation of alveolar epithelial cells in 6/6 patients. In nonpulmonary tissues, ACE2 and TMPRSS2 were both expressed in biliary duct epithelium, renal distal tubule and collecting duct, prostatic acinar cells, and glandular epithelium of small intestine. In addition, ACE2 mRNA were also detected in endothelial cells within lymph node, enterocytes lining the crypts, and glands of the intestine and myocardial endothelial and stromal cells. (Supplementary Fig. 3, Supplementary Fig. 4) The expression of AR, the transcriptional regulator of TMPRSS2, were also observed in the bronchial, bronchiolar and alveolar epithelial cells, subepithelial bronchial, and tracheal sero-mucinous glands of the pulmonary tissues, as well as nonpulmonary tissues including the prostate,

thyroid, kidney, and liver (Supplementary Fig. 5). It is important to mention that SARS-CoV-2 infection was detected in most tissues and cell types expressing ACE2 and TMPRSS2, except adrenal, pancreas, stomach, esophagus, and small intestine, as described above. SARS-CoV-2 viral signals were also detected in tissues where ACE2 and TMPRSS2 transcripts were not detected, including spleen, colon, heart (ACE2 only), lymph node (ACE2 only), and uterus (ACE2 only).

## Discussion

The highly contagious and rapidly spreading COVID-19 has evolved into a global health threat in 2020. This disease is now known to manifest with a wide spectrum of severity with severe disease resulting in intense and multiorgan dysfunction[13,14]. Currently, there is a limited understanding of the pathophysiology of COVID-19, and no targeted therapy is yet available for treatment of established SARS-CoV-2 infection. As a follow-up to our recent study on transcriptional regulation of SARS-CoV-2 entry factors in lung[8], herein, we studied the tissue distribution and localization of SARS-CoV-2 virus, viral replication, the transcripts of the host cell entry factors, ACE2 and TMPRSS2, and the transcriptional regulator of TMPRSS2, AR, in postmortem human organ systems using in situ and qRT-PCR approaches, and correlate those results to the histopathologic findings in COVID-19 patient autopsy cases.

Clinical autopsy examination revealed ARDS due to SARS-CoV-2 infection to be implicated in the demise of all six patients. The medical history in this series was enriched for at least one underlying or predisposing medical condition in each of the six patients including diabetes, hypertension, coronary artery disease, asthma, and obesity. These findings are consistent with previous reports in the literature suggesting older age (>60 years), serious co-morbidities, and tobacco exposure to be associated with a

higher risk of developing ARDS and mortality in COVID-19 patients, especially in males[13].

Histologic examination of pulmonary tissues revealed acute diffuse alveolar damage with prominent hyaline membranes as a common COVID-19 pathological feature, in concordance with the literature[1,15–17]. In our study, the most common pathological changes encountered in nonpulmonary tissues included myofibrillary hypertrophy in the heart, tubular injury in the kidney, and steatosis in the liver. These observations are in line with published reports that SARS-CoV-2 infection may cause acute injury in heart, kidney, and other tissues through a direct cytopathic effect[14,18]. In addition, catecholamine surge leads to cytokine storm imploding as systemic inflammatory response syndrome (SIRS), leading to multiorgan failure, which has been documented as a major cause of death in COVID-19 patients[1,14].

We further employed a single-molecule RNA-ISH technology for sensitive and specific detection of SARS-CoV-2 in FFPE tissue sections. This technology has been adapted to facilitate SARS-CoV-2 diagnosis and research, and has demonstrated SARS-CoV2-2 virus detection in cultured cells[19–21], rodent[22], non-human primates[23,24], and human tissues autopsies[25–29]. Employing RNA-ISH, we were able to concurrently document both the tissue distribution of SARS-CoV-2 viral particles and the host entry factors in pulmonary and extrapulmonary tissues. The SARS-CoV-2 RNA-ISH staining pattern and signal localization resembles those of the SARS-CoV-2 IHC signals. Our observations of RNA-ISH staining patterns in pulmonary tissues are consistent with published studies[28,30] and the viral localization matches the pulmonary histopathological findings in our study and other postmortem and retrospective studies[16,25,26] Among nonpulmonary tissues, very rare signal clusters were observed in kidney, heart, liver, thyroid, spleen, prostate, uterus, and colon tissue sections, but no individual signal dots were detected. Low viral signals were detected in these specimens by qRT-PCR as indicated by the high Ct values in the range of 27.4–38.8. The failure to detect SARS-CoV-2 viral particles by RNA-ISH might be due to the low viral count and/or RNA degradation prior to tissue fixation.

It is noteworthy that apart from pulmonary infection, there has been limited experimental evidence of SARS-CoV-2 infection in nonpulmonary tissues previously[1,31]. A few research groups have attempted by means of RT-PCR, electron microscopy or IHC, to detect SARS-CoV-2 viruses in nonpulmonary tissues and have recorded the presence of viral particles in heart[27,32], kidney[16,32], liver[33,34], placenta[35–37], gastrointestinal tract[38], and skin[39]. SARS-CoV-2 viruses were also detected in kidney[16,29] and placenta[36,37,40] by RNA-ISH. The current study is by far the most comprehensive report of SARS-CoV-2 infection in multiorgan systems and the first report of SARS-CoV-2 viral detection in lymph node, spleen, thyroid, colon, prostate, and uterus. The identification of SARS-CoV-2 viral infection in distant, non-pulmonary tissues, like prostate and uterus, suggest the wide spread of SARS-CoV-2 in the diverse human organ systems.

The SARS-CoV-2 host entry mediator, TMPRSS2, was first found to be highly enriched in prostate cancer[41] were reported in the prostate of six male COVID-19 patients[33]. Recent single-cell RNA expression studies mapped SARS-CoV-2 entry factors in a broad range of human tissues and predicted that testicular spermatogonial cells and prostatic endocrine cells are susceptible to SARS-CoV-2 infection[8,42], and we presented evidence of SARS-CoV-2 infection in prostatic tissue. Overall, our examination of SARS-Cov-2 tissue localization by in situ and qRT-PCR approaches revealed a systemic fashion of viral infection, which may explain the multisystemic involvement observed in COVID-19 disease.

The 2003 SARS coronavirus and 2019 SARS-CoV-2 virus employ host proteins ACE2 and TMPRSS2 to gain cell entry. The viral spike (S) glycoprotein recognizes and binds to ACE2 receptor on host cells and is cleaved and activated by cell surface transmembrane protease TMPRSS2[6,43]. Targeting the transcriptional regulation or activity of these host factors could inhibit SARS-CoV-2 infection and replication. TMPRSS2 has been widely studied in prostate cancer where it is highly expressed in an androgen-dependent manner[44]. The recurrent oncogenic TMPRSS2-ETS gene fusions are also found in more than 50% of prostate cancers[41]. AR inhibitors have been developed for treatment of prostate cancer and could be repurposed for COVID-19[45]. In our recent study, we demonstrated that ACE2, TMPRSS2 and AR are co-expressed in a subset of lung epithelial cells under the transcriptional regulation of androgen; and transcriptional repression of AR inhibited SARS-CoV-2 infection in vitro[8]. However, the distribution of ACE2 and TMPRSS2 expression in human organ systems associated with viral infection has not been comprehensively characterized in the literature. Herein, we further characterized the tissue distribution of ACE2, TMPRSS2, and AR transcripts in multiorgan systems and found that the distribution of host entry factors, ACE2 and TMPRSS2, and regulator, AR, highly overlaps with the viral infection sites in pulmonary tissues and numerous nonpulmonary tissues, supporting the notion that targeting ACE2 and/or TMPRSS2 could be an effective treatment strategy to counter SARS-CoV-2 infection from a multiorgan perspective and inhibition of AR would be effective against SARS-CoV-2 infection[8]. The expression and tissue distribution of entry and attachment receptors can influence viral tropism and pathogenicity. Our study provides a panoramic view of such factors at a systemic level.

We also investigated viral replication activity by targeting the minus strand of SARS-CoV-2. Active viral replication events were detected in lung, bronchus, and trachea. The detection frequency of viral replication is much less than that of viral particle signals in the tissues mentioned above. Similar observation has been reported in lung tissue[28]. Replication events and kinetics may influence viral pathogenicity, tissue tropism, and accelerated clinical organ system involvement and decline. Our study documents such events at a systemic level in COVID-19 disease. Such findings also provide valuable resources for the development of SARS-CoV-2 replication inhibitors for disease management.

Based on our observations, COVID-19 disease demonstrates heterogeneity with respect to viral load and viral cytopathic effects among various organ systems, even within the same patient. The highest viral load was observed in lung tissues and concentrated within the lung alveolar hyaline membrane and intra-alveolar spaces, followed by bronchus and trachea in the respiratory system. It is currently unclear whether heterogeneous viral abundance and distribution is associated with disease severity.

Our study cohort included a COVID-19 patient with prior kidney transplant receiving immunosuppression therapy. The SARS-CoV-2 RNA-ISH staining revealed extremely high viral infection in the lung tissue of this decedent along with SARS-CoV-2 viral infection and viral inflammatory changes in the transplant kidney, liver, and uterus. Recently studies linked high viral load with disease severity and transmissibility. A hospital-based study in China revealed that the severe COVID-19 cases tend to have higher viral load and longer virus-shedding period than mild cases[46]. An observational study conducted in India showed that the index cases with high viral load detected by RT-PCR transmitted 6.25 secondary cases on average whereas the cases with low viral load transmitted an average of 0.8 case[47]. Immunocompromised persons may experience persistent SARS-CoV-2 infection and bear accelerated viral evolution under immunocompromised state[48]. Therefore, the high viral load observed in the COVID-19 patient under immunosuppression therapy could be related to the immune suppression status and

SARS-CoV-2 infected, immunosuppressed patients may act as super-spreaders under certain circumstances.

Overall, our findings show co-existence of SARS-CoV-2 infection and host entry factors in multiple pulmonary and nonpulmonary tissues. We believe that a detailed characterization of such biomarkers in conjunction with histopathological assessment to assess disease severity and progression will guide future coronavirus biology studies on patients with advanced disease, as well as provide a framework for identifying better, novel or specific antiviral therapeutics.

### Data availability

The RT-PCR data are available in the supplementary information files. All other data that support the findings of this study are available from the corresponding author upon reasonable request.

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

## Acknowledgements

We thank the patients and their families for participation in the autopsy program at the University of Michigan Health System. They further thank Paul Harms, M.D., and Angela Wilson for helpful suggestions, Monique Micallef for assisting the autopsies, and the histology staff of the Department of Pathology at the University of Michigan Health System. We are thankful to Jyoti Athanikar for assistance with manuscript preparation and submission. This work was supported by the following: Prostate Specialized Programs of Research Excellence Grant P50-CA186786, National Cancer Institute Outstanding Investigator Award R35-CA231996, National Cancer Institute P30-CA046592, and COVID-19 Administrative Supplement to this grant. A.M.C. is a Howard Hughes Medical Institute Investigator, A. Alfred Taubman Scholar, and American Cancer Society Professor.

## Author contributions

X.M.W., R. Mannan, C.F., J.L.M., L.P., A.M.C. and R. Mehra designed research; X.M.W., R. Mannan, L.X., L.M., F.S., R.W., S.Z.W., Y.Z. and X.C. performed research; J.J. and A.W. performed autopsy and generated clinical report; X.M.W., R. Mannan, L.X., Y.Q., Y.Z. and R. Mehra analyzed data; X.M.W., R. Mannan, L.X., E.A. and R. Mehra wrote the paper.

## Competing interests

The authors declare no competing interests.
