## [Peer Review File · Communications Medicine]

Reviewers' comments:

Reviewer #1 (Remarks to the Author):

Working with tissue specimens from 6 COVID-19 autopsy cases, the authors examined tissue distribution of SARS-CoV-2 virus, viral replication, and host entry factors (ACE2 and TMPRSS1) in various anatomical sites using RNA in-situ hybridization (RNA-ISH) and quantitative reverse transcription polymerase chain reaction (qRT-PCR). They detected SARS-CoV-2 virus RNA and viral replication-form RNA in pulmonary tissues by RNA-ISH and a variety of non-pulmonary tissues including kidney, heart, liver, spleen, thyroid, lymph node, prostate, uterus, and colon by qRT-PCR. Heterogeneity in viral load and viral cytopathic effects was observed among various organs, among individuals and within the same patient; they also observed unusually high viral load in lung tissue by both RNA-ISH and qRT-PCR in a patient under immunosuppressant therapy. ACE2 and TMPRSS2 expression was found to overlap with the infection sites.

Thus far, SARS-CoV-2 and its disease have been extensively studied, including its pathogenesis at tissue, cellular, and molecular levels. The expression and distribution of ACE2, TMPRSS2, and localization of the virus have also been reported several other groups. To this, the results as reported in the current paper provides confirmation to previous studies. Histological findings described also match that of previous autopsy series.

The study is well designed and data well organized. There are, however, several areas that need to be further addressed to strengthen the study.

This group of investigators had recently published a very interesting study revealing co-expression of the androgen receptor (AR), TMPRSS2, and ACE2 in bronchial and alveolar cells in lung tissues and demonstrated that transcriptional repression of the AR enhanceosome by AR antagonists inhibited SARS-CoV2 infection in vitro. It is important therefore to detect androgen receptors in the current study, along with the virus, ACE2 and TMPRSS2. This will provide much confirmation of their previous study and confer uniqueness to the current paper, as compare to what already reported in literature.

Line 185: "RNA-ISH signals were observed to be patchy and mostly extracellular," and Figure 1. Can the authors clarify as to the specificity of the staining, since by common sense cell-free viral particles are very rare in human tissues (as opposed to laboratory setting such as cell culture fluid)? This reviewer understands that a negative and an influenza lung tissue specimens were used as negative controls. However, a "nonsense" probe control is also necessary in such a situation when the location of positive signal turned out to be unconventional. In addition, as shown in Figure 1B, with such strong and extensive RNA signal, there must be abundant viral proteins as well, since protein is much more abundant in viral particles than RNA. Thus, a confirmatory detection using immunohistochemical staining in the same tissue sections should be able to provide convincing validation, and should be performed in this study.

The use of a SARS-CoV-2-S-sense probe to detect a replication form of viral RNA is intriguing and the findings are quite interesting and novel. However, from Figure 2A-C, the clusters of brown signal were seen mostly in extracellular area (in debris), making it more important to validate its specificity, since no viral replication can occur out of cells. Only Fig 2D showed intracellular localization.

The usage of “in vitro” and “in vivo” should be consistent. It was used in line 86 as in vitro, but in line 308 as in vivo, referring to the same experiment. Please clarify.

Reviewer #2 (Remarks to the Author):

Wang and colleagues present a comprehensive study addressing tissue distribution and localization of SARS-CoV-2, viral replication and host entry factors, using human autopsy samples and conventional histopathology, RNA-ISH and qRT-PCR. SARS-CoV-2 was detected in the lung, kidneys, heart, liver, spleen, thyroid, lymph nodes, the prostate, uterus and the colon. ACE2 and TMPRSS2 expression overlapped with the infection sites. Depending on the immune status of the respective patients, heterogeneity in viral load, viral cytopathic effects and between individuals and within the same patient was reported.

The manuscript is concisely written, the m&M as well as the results part can easily be understood and the illustrations are adequate.

There are, however, some points which the authors need to address:

1. Supplementary Table 3. SARS-CoV-2 qRT-PCR and RNA-ISH result comparison. Supplementary Table 3 is quite important, considering that it addresses one of the main points of the manuscript ◊ the comparison of different mechanisms of detection. The table should be restructured and address the individual patients in a more comprehensive manner (easier to interpret by the reader)
2. The study includes only six patients. The claims of the authors regarding host entry factors make sense, especially considering current literature. Albeit, based on such a small study number, the authors should rephrase their postulations, particularly in regard to transplant recipients
3. While the danger of immunosuppressants and other host factors are discussed, the potential role(s) of vaccination is not. The authors should at least briefly include this in the discussion
4. The authors should combine their in-situ approach with immunohistochemistry e. g. staining against nucleocapsid proteins of SARS-CoV-2; complementary detection / co-detection of virus proteins (or lack thereof) in a compartment-specific manner will add significant value to the manuscript.

Manuscript ID: COMMSMED-21-0118

Manuscript Title: Characterization of SARS-CoV-2 virus infection, viral replication, and host entry factors in a COVID-19 autopsy series

>> Thank you very much for reviewing the manuscript and providing valuable comments to further improve the relevance and impact of this study.

>> We have performed additional experiments and attempted to address all of the reviewers' and editors' comments and suggestions as detailed below in this document. To highlight some salient features, in response to the concerns regarding the specificity of viral detection, we utilized several and multi-platform experimental approaches to confirm the findings, including **1)** duplex RNA-ISH for co-detection of SARS-CoV-2 S and N genes, **2)** duplex RNA-ISH for co-detection of virus and viral replication, **3)** RNase treatment to confirm the detection of RNA signals/specificity, **4)** detection of SARS-CoV-2 virus by immunohistochemistry (IHC), and **5)** detection of negative strand of SARS-CoV-2 virus by strand-specific RT-PCR, as additional evidence of viral replication. We have now also reported the expression of *AR* transcripts in this autopsy series.

We have made appropriate changes to the manuscripts following reviewers' suggestions. We believe that this revised manuscript with its observations now provide a greater and more factual insight into the pathobiology of COVID-19 disease, and hope it will be considered towards publication.

>> Our point-by-point response is listed below.

Authors' Point by Point Response:

Reviewer #1 (Remarks to the Author):

Working with tissue specimens from 6 COVID-19 autopsy cases, the authors examined tissue distribution of SARS-CoV-2 virus, viral replication, and host entry factors (ACE2 and TMPRSS1) in various anatomical sites using RNA in-situ hybridization (RNA-ISH) and quantitative reverse transcription polymerase chain reaction (qRT-PCR). They detected SARS-CoV-2 virus RNA and viral replication-form RNA in pulmonary tissues by RNA-ISH and a variety of non-pulmonary tissues including kidney, heart, liver, spleen, thyroid, lymph node, prostate, uterus, and colon by qRT-PCR. Heterogeneity in viral load and viral cytopathic effects was observed among various organs, among individuals and within the same patient; they also observed unusually high viral load in lung tissue by both RNA-ISH and qRT-PCR in a patient under immunosuppressant therapy. ACE2 and TMPRSS2 expression was found to overlap with the infection sites.

>> Thank you for providing a summary of our approach and findings.

Thus far, SARS-CoV-2 and its disease have been extensively studied, including its pathogenesis at tissue, cellular, and molecular levels. The expression and distribution of ACE2, TMPRSS2, and localization of the virus have also been reported several other

groups. To this, the results as reported in the current paper provides confirmation to previous studies. Histological findings described also match that of previous autopsy series.

The study is well designed and data well organized. There are, however, several areas that need to be further addressed to strengthen the study.

1. This group of investigators had recently published a very interesting study revealing co-expression of the androgen receptor (AR), *TMPRSS2*, and *ACE2* in bronchial and alveolar cells in lung tissues and demonstrated that transcriptional repression of the AR enhanceosome by AR antagonists inhibited SARS-CoV2 infection in vitro. It is important therefore to detect androgen receptors in the current study, along with the virus, *ACE2* and *TMPRSS2*. This will provide much confirmation of their previous study and confer uniqueness to the current paper, as compare to what already reported in literature.

>> Thank you for this excellent suggestion of including the detection of androgen receptor (AR) in the COVID-19 autopsy cohort. To address this comment, we performed *AR* RNA-ISH on the tissue samples in our autopsy cohort and observed *AR* transcript signals in epithelial cells of the pulmonary tissues, as well as non-pulmonary tissues including the prostate, thyroid, kidney and liver. The localization of *AR* expression matches with that of *TMPRSS2* expression and the SARS-CoV-2 infection sites. The *AR* RNA-ISH data is summarized in **Supplementary Figure 5** and the findings presented within the Results section (**Page 11, lines 268-271**).

2. Line 185: “RNA-ISH signals were observed to be patchy and mostly extracellular,” and Figure 1. Can the authors clarify as to the specificity of the staining, since by common sense cell-free viral particles are very rare in human tissues (as opposed to laboratory setting such as cell culture fluid)? This reviewer understands that a negative and an influenza lung tissue specimens were used as negative controls. However, a “nonsense” probe control is also necessary in such a situation when the location of positive signal turned out to be unconventional. In addition, as shown in Figure 1B, with such strong and extensive RNA signal, there must be abundant viral proteins as well, since protein is much more abundant in viral particles than RNA. Thus, a confirmatory detection using immunohistochemical staining in the same tissue sections should be able to provide convincing validation, and should be performed in this study.

>> We thank the reviewer for this suggestion as the specificity of the RNA-ISH assay is crucial to this study.

The SARS-CoV-2 virus RNA-ISH signals were observed intracellularly in the alveolar epithelium and extracellularly within the hyaline membranes which are composed of secretory proteins and dead cells lining the alveoli. To be more precise, we have changed the description from “patchy and mostly extracellular” to “within the hyaline membranes lining the alveoli” (**Page 8, line 202**).

We further evaluated the specificity of RNA-ISH utilizing several aspects and diverse testing platforms as listed below-

1. Besides the negative control samples, like normal lung tissue and influenza lung tissue specimens, we also employed negative control probe (DapB) staining for every tissue specimen included in this study to be able to monitor the assay background, as mentioned in the Methods section (**Page 5, lines 121-122**). Examples of DapB stainings were shown in **Supplementary Figure 1 I&J**.
2. Based on the reviewer's suggestion, we performed SARS-CoV-2 nucleocapsid antibody IHC on SARS-CoV-2 RNA-ISH positive autopsy tissue samples and RNA-ISH negative normal control lung samples (**Supplementary Figure 2**). We observed similar staining patterns and signal locations between the RNA-ISH and IHC results and across assays, with parallel negative results in the normal (uninfected) control lung sample. These data were described in the Results section (**Page 9, lines 223-227**).
3. We further confirmed the specificity of SARS-CoV-2 virus RNA-ISH signals by co-detection of SARS-CoV-2 spike (S) gene (both positive and negative strand) and nucleocapsid (N) gene using a duplex RNA-ISH strategy (**Figure 3**). Based on this assay results, we observed co-expression of S (green) and N (red) gene dominantly in the hyaline membranes and intra-alveolar septa of lung. The observation of S/N co-detection in an in-situ fashion is another piece of evidence supporting the RNA-ISH signal specificity. These data were described in the Results section (**Page 8-9, lines 212-218**).
4. We also performed additional confirmatory experiments by RNase treatment to confirm that the observed RNA-ISH signals truly emanate from RNA targets. Upon RNase treatment, no signals were detectable with probes against SARS-CoV-2 S positive (**Supplementary Figure 1 C&D**) or negative strand (**Supplementary Figure 1 G&H**) in the same tissue samples where positive signals were detected without the RNase treatment (**Supplementary Figure 1 A&B, E&F**). These data were described in the Results section (**Page 9, lines 219-222**).

In addition, Massoth, L.R. et al.* reported the comparison of RNA-ISH and IHC techniques for the detection of SARS-CoV-2 in human tissues, where the authors reported 100% specificity of RNA-ISH from the same vendor, and 53.3% specificity of IHC in 19 lung tissues collected from 8 COVID-19 patients and negative control lung tissues from 37 individuals collected prior to the pandemic.

*Massoth, L.R. et al. Comparison of RNA In Situ Hybridization and Immunohistochemistry Techniques for the Detection and Localization of SARS-CoV-2 in Human Tissues. *Am J Surg Pathol* (2020).

3. The use of a SARS-CoV-2-S-sense probe to detect a replication form of viral RNA is intriguing and the findings are quite interesting and novel. However, from Figure 2A-C, the clusters of brown signal were seen mostly in extracellular area (in debris), making it more important to validate its specificity, since no viral replication can occur out of cells. Only Fig 2D showed intracellular localization.

>> Thank you for these comments. We took the reviewer's suggestion and updated the images in **Figure 2**. As mentioned above, we performed duplex RNA-ISH of SARS-CoV-2-S-sense probe

and N probe and observed co-detection of both signals to support the signal specificity (**Figure 3 C&D**), beside the control probes' staining and RNase treatment evaluated in this study.

To further confirm the signal specificity of the viral replication detected by RNA-ISH, we performed strand-specific RT-PCR for detection of the minus strand SARS-CoV-2 viral RNA or the positive strand SARS-CoV-2 viral RNA using the method developed by Hogan C.A. et al.[#] The strand-specific RT-PCR result is consistent to the RNA-ISH result (**Supplementary Table 4**) for both SARS-CoV-2 S-sense positive (replication detected) and negative (replication not detected) samples. These data were described in the Results section (**Page 9, lines 235-238**).

[#]Hogan, C.A. et al. Strand-Specific Reverse Transcription PCR for Detection of Replicating SARS-CoV-2. *Emerg Infect Dis.* 2021;27(2):632-635. <https://doi.org/10.3201/eid2702.204168>

4. The usage of “in vitro” and “in vivo” should be consistent. It was used in line 86 as in vitro, but in line 308 as in vivo, referring to the same experiment. Please clarify.

>> We thank the reviewer for this suggestion. We have changed the statement in **Page 13 Line 341** to “*in vitro*” for consistency.

Reviewer #2 (Remarks to the Author):

Wang and colleagues present a comprehensive study addressing tissue distribution and localization of SARS-CoV-2, viral replication and host entry factors, using human autopsy samples and conventional histopathology, RNA-ISH and qRT-PCR. SARS-CoV-2 was detected in the lung, kidneys, heart, liver, spleen, thyroid, lymph nodes, the prostate, uterus and the colon. ACE2 and TMPRSS2 expression overlapped with the infection sites. Depending on the immune status of the respective patients, heterogeneity in viral load, viral cytopathic effects and between individuals and within the same patient was reported.

The manuscript is concisely written, the m&M as well as the results part can easily be understood and the illustrations are adequate.

>> Thank you for providing a summary of our approach and findings.

There are, however, some points which the authors need to address:

1. Supplementary Table 3. SARS-CoV-2 qRT-PCR and RNA-ISH result comparison. Supplementary Table 3 is quite important, considering that it addresses one of the main points of the manuscript à the comparison of different mechanisms of detection. The table should be restructured and address the individual patients in a more comprehensive manner (easier to interpret by the reader)

>> We thank the reviewer for this suggestion. We restructured **Supplementary Table 3** to present the qRT-PCR and RNA-ISH side by side for each individual patient separately.

2. The study includes only six patients. The claims of the authors regarding host entry

factors make sense, especially considering current literature. Albeit, based on such a small study number, the authors should rephrase their postulations, particularly in regard to transplant recipients

>> Thank you for pointing this out. We revised the discussion and conclusion regarding the results related to the transplant recipients. We focused on describing the RNA-ISH staining observed in this particular patient to report the very high viral infection in the lung tissues of this individual and extended to recent studies linking the high viral load with disease severity and transmission. We also removed the postulation regarding the immunosuppression status and super spreader events from the Discussion section.

3. While the danger of immunosuppressants and other host factors are discussed, the potential role(s) of vaccination is not. The authors should at least briefly include this in the discussion.

>> Thank you for the suggestion. Vaccination is now a proven and critical tool to help stop the spread of the COVID-19 disease. Unfortunately, this study was conducted prior to the FDA approval of emergency usage of the COVID-19 mRNA vaccines and hence beyond the scope of this current study. We included a statement in the Methods section (**Page 4, lines 101-102**) stating that ‘this study was conducted prior to the FDA approval of emergency use of COVID-19 vaccines and none of the patients in this study received COVID-19 vaccination’.

4. The authors should combine their in-situ approach with immunohistochemistry e. g. staining against nucleocapsid proteins of SARS-CoV-2; complementary detection / co-detection of virus proteins (or lack thereof) in a compartment-specific manner will add significant value to the manuscript.

>> We agree and thank you for these suggestions to further strength this study. We addressed both reviewers’ comments by performing the duplex RNA-ISH and IHC approaches. As detailed in the reply to reviewer 1’s 2nd and 3rd comment, we performed co-staining of SARS-CoV-2 nucleocapsid gene and spike gene (either plus or minus strand) by duplex RNA-ISH and observed co-detection of both gene signals in the same anatomic locations as single-plex RNA-ISH and with a similar staining pattern (**Figure 3**). We also performed nucleocapsid IHC in the SARS-CoV-2 RNA-ISH positive tissue samples and negative normal control lung samples. Again, the IHC staining pattern and signal localization match the RNA-ISH results (**Supplementary Figure 2**).

REVIEWERS' COMMENTS:

Reviewer #1 (Remarks to the Author):

All questions addressed sufficiently.

Reviewer #2 (Remarks to the Author):

Wang and colleagues present a comprehensive study addressing tissue distribution and localization of SARS-CoV-2, viral replication and host entry factors.

All my queries have been addressed; I have no further comments to make and recommend this important manuscript for publication. My congratulations to the authors for their concise work.